# A Survey in Hawaii for Parasitoids of Citrus Whiteflies (Hemiptera: Aleyrodidae), for Introduction into Greece

**DOI:** 10.3390/insects14110858

**Published:** 2023-11-04

**Authors:** Maria-Vasiliki Giakoumaki, Panagiotis Milonas, Spyridon Antonatos, Vasiliki Evangelοu, George Partsinevelos, Dimitrios Papachristos, Mohsen M. Ramadan

**Affiliations:** 1Scientific Directorate of Entomology and Agricultural Zoology, Benaki Phytopathological Institute, 14561 Kifissia, Greece; m.giakoumaki@bpi.gr (M.-V.G.); p.milonas@bpi.gr (P.M.); s.antonatos@bpi.gr (S.A.); v.evangelou@bpi.gr (V.E.); g.partsinevelos@bpi.gr (G.P.);; 2Division of Plant Industry, Hawaii Department of Agriculture, Honolulu, HI 96814, USA

**Keywords:** Hymenoptera, natural enemies, citrus, citrus blackfly, orange spiny whitefly, *Encarsia perplexa*, *Amitus hesperidum*

## Abstract

**Simple Summary:**

The orange spiny whitefly, *Aleurocanthus spiniferus*, has invaded Greece and expanded its distribution in other countries in the European Union since 2008. It is a polyphagous pest that includes several highly important crop plants, such as citrus. Damage symptoms and effects on plants are identical to those of the citrus blackfly, *Aleurocanthus woglumi* (both Hemiptera: Aleyrodidae). The species also share a complex of natural enemies that were successfully introduced to Hawaii during 1974–1998. A short expedition to the islands of Oahu, Hawaii, and Kauai was conducted to retrieve the prominent natural enemies for introduction into Greece. Hawaii was chosen because it does not have citrus diseases and because of the existence of connections to facilitate research and export permits. Infested leaves were shipped to a quarantine facility in Greece for parasitoid emergence and evaluation. The identity of emerged parasitoids and host testing on the orange-spiny whitefly Greece strain were conducted. Only one parasitoid was abundant, characterized using molecular analysis as *Encarsis perplexa*. A summary of infestation records, parasitism rates, and localities on the Hawaiian Islands has been reported here for the first time since the release of parasitoids. Results showed that the infestations of citrus trees were minimal on the islands of Hawaii and Oahu, primarily on pummelo and sweet orange. Citrus whiteflies were not detected on the island of Kauai during this survey. *E. perplexa* had parasitism rates ranging from 0 to 28% on the island of Hawaii and 11 to 65% on the island of Oahu. A starter colony of the parasitoids has been colonized in the Greece Quarantine Facility for evaluation. This was the first field survey of Hawaii since the introduction and release of citrus whitefly natural enemies. Further surveys should be repeated in different countries to eliminate the risk of disease introduction.

**Abstract:**

Whitefly species of *Aleurocanthus spiniferus* (Quaintance) and *A. woglumi* Ashby (Hemiptera: Aleyrodidae) are serious pests of citrus and other important fruit crops. The problem of citrus has initiated the successful introduction of several natural enemies for biocontrol programs in Hawaii and many other countries. Here, we summarized the history of infestation and biocontrol efforts of the two whiteflies in Hawaii for possible parasitoid importation into Greece. Two Platygasteridae (*Amitus hesperidum* Silvestri, *A. spiniferus* (Brethes), and three Aphelinidae (*Encarsia clypealis* (Silvestri), *E. smithi* (Silvestri), *E. perplexa* Huang, and Polaszek) were released in Hawaii for biocontrol of the citrus whiteflies during the period 1974–1999. The aphelinid *Cales noacki* Howard, purposely released for *Aleurothrixus flococcus* (Maskell) in 1982, was also reported to attack other whiteflies, including *Aleurocanthus* species, on citrus. An additional aphelinid parasitoid, *Encarsia nipponica* Silvestri, native to Japan and China, was accidentally introduced and found to attack both citrus whiteflies on the islands. Since the colonization of introduced parasitoids in infested fields on four Hawaiian Islands, no survey has been conducted to evaluate their potential impact. We conducted two short surveys during September–November 2022 on the islands of Kauai, Hawaii, and Oahu to introduce the dominant parasitoids to Greece for the biocontrol of *A. spiniferus*. Results showed that the infestation level was very low on Kauai, Hawaii, and Oahu Islands, with a mean infestation level range of 1.4–3.1 on Hawaii and Oahu Islands, mostly on pummelo and sweet orange, with no detection on the island of Kauai. The dominant parasitoid was characterized as *Encarsia perplexa*, using molecular analysis. Its parasitism rates ranged from 0 to 28% on the island of Hawaii and 11 to 65% on the island of Oahu. Emerged parasitoids have been reared in Greece for evaluation. This was the first field survey of Hawaii since the introduction and release of citrus whitefly natural enemies.

## 1. Introduction

The orange spiny whitefly, *Aleurocanthus spiniferus* (Quaintance), and citrus blackfly, *A. woglumi* Ashby (Hemiptera: Aleyrodidae), are two of the most important and hazardous citrus pests [1,2]. Both species are regulated as quarantine pests for the European Union [3]. *Aleurocanthus woglumi* has not invaded Europe yet. However, climatic modeling predicts that areas in the Mediterranean basin are suitable for establishment [4]. *Aleurocanthus spiniferus* has already invaded Europe, with the first record in Italy in 2008 [5]. Since then, its distribution in the European Union has expanded, including Albania [6], Croatia [7], Greece [1], and Montenegro [8]. Both species cause similar symptoms to the infested host plants. In Hawaii, during a heavily infested period in 1998, infested trees were not fruiting, with a range of 50–>600 citrus blackfly pupae per infested leaf ([2], Ramadan, Hawaii Department of Agriculture (HDOA) unpublished record). The main hosts are citrus species, but they are quite polyphagous, with a wide host range that includes mangoes and coffee [9]. Direct damage is caused by nymphs that suck the sap from the leaves, causing a lack of nutrients and water. Additionally, nymphs excrete honeydew drops that fall on the leaves, where the growth of sooty mold is promoted, which eventually covers the whole upper surface of the infested leaves. Both *A. spiniferus* and *A. woglumi* originate from Southeastern Asia [4,10]. In their native area, they are not considered major pests; however, during the 20th century, both species invaded other parts of the world, causing substantial damage, mainly to citrus crops [2,3,4]. Following their accidental introduction to new areas, biological control programs were initiated to mitigate the damage caused by the outbreaks of the two whiteflies [2,10,11,12,13,14].

In the early seventies, *A. spiniferus* was introduced in Hawaii on rose foliage in Honolulu, Oahu Island, in 1974. Subsequent surveys discovered it on navel orange, lime, tangerine, and pear, but infestations were reportedly low [15]. Two decades later, *A. woglumi* was also found being introduced on the main Hawaiian Islands [15,16]. In both cases, the introduction of citrus whitefly species followed the introduction of biological control agents to compact their population outbreaks. Specifically, the parasitoid *Encarsia smithi* Silvestri and *E. clypealis* (Silvestri) (Hymenoptera: Aphelinidae) were introduced in the islands of Oahu and Hawaii [17] to control *A. spiniferus*, and *E. perplexa* Huang and Polaszek (Hymenoptera, Aphelinidae) (misidentified at that time as *Encarsia opulenta* Silvestri) and *Amitus hesperidum* Silvestri (Hymenoptera: Platygastridae) were introduced to control *A. woglumi* [16], Table 1. *Encarsia smithi* was introduced to the island of Oahu from Japan in 1974, among three other Aphelinid species that were found to naturally parasitize *A. spiniferus* in Oahu. *Encarsia smithi* was the most important species, leading to a sufficient reduction of the *A. spiniferus* population [17]. It was also reported on *A. woglumi* (Table 1). In 1998, exploratory investigations were carried out in Central America in Guatemala, and two parasitoid species, *E. perplexa* and *A. hesperidum*, were introduced to Hawaii. The wasps were mass-reared on Oahu and released on other islands (Table 1). The two wasp species managed to disperse naturally and successfully controlled the population of *A. woglumi* in all releasing sites (Table 1) [18]. Another parasitoid, *E. nipponica* Silvestri, native to Japan and China, was discovered accidentally as a parasitoid of both whiteflies with low parasitism ranging from 0 to 11.5% (*n* = 103 infested leaves, Ramadan unpublished data). Similarly, the aphelind parasitoid, *Cales noacki* Howard, was released in 1981 for biological control of *Aleurothrixus floccosus* (Maskell), subsequently reported to attack *Aleurocanthus* species on citrus [19], HDOA records. There has been no field evaluation since the introduction and release of all the parasitoids, but citrus growers on the islands were content, and less chemical control was used in their fields [2].

The recent invasion of *A. spiniferus* in the European Union and Greece, in particular, has caused the initiation of a new biological control program against this pest by introducing exotic natural enemies. Moreover, since the introduction of these parasitoids in Hawaii, no surveys have been conducted to determine the status of *A. woglumi* or *A. spiniferus* in the islands and the presence and abundance of their introduced parasitoids. Thus, an exploratory investigation on the Hawaiian Islands was performed to investigate the current parasitism rate of *A. woglumi* and to identify the parasitoid assembly species that were present in the populations of the pests. The results of those short surveys will provide crucial information to examine the possibility of a successful introduction of those parasitoids, either in Greece or in any other area where those serious pests have been established.

## 2. Materials and Methods

### 2.1. Collection and Emergence of Parasitoids

Two surveys were performed in September and November 2022 on the islands of Hawaii, Kauai, and Oahu (Figure 1). A total of 62 sites located on the three Hawaiian Islands were surveyed (Hawaii Island with the GPS coordinates of 19°44′30.3180″ N, 155°50′39.9732″ W; Oahu Island GPS coordinates of 21°18′56.1708″ N, 157°51′29.1348″ W; Kauai Island GPS coordinates of 22°6′30.7548″ N, 159°29′48.3540″ W (https://www.latlong.net (accessed on accessed on 25 October 2023)).

Host plants (mainly citrus trees) were macroscopically surveyed for citrus whitefly infestation, and infested leaves were collected (Table 2 and Table 3). Citrus whiteflies have been very rare to be found in Hawaii in recent years. Sites were selected as orchards of citrus, trees in agriculture experimental stations, and local residential homes were examined with the permission of landlords. Sites were selected randomly for the survey team to enter property everywhere on the visited islands. Based on the presence of citrus trees, the survey team searched the trees for infestations and obtained permission to clip infested branches or leaves. Leaves with mature whiteflies were picked, not those with only eggs or small nymphs that would not produce parasitoids.

The infested leaves were placed in paper envelopes, which were sealed (AJM paper grocery, lunch bags, USA, of different sizes as needed for holding infested leaves). Plastic bags are not suitable for live insects, https://www.gofoodservice.com/brand/ajm (accessed on 10 September 2023). Infested leaves were picked according to the stage of the nymphs of the citrus blackfly. Leaves that carried all the stages and older nymphs were preferred. Leaves with mature nymphs (third and fourth nymphs) were preferred for collection. Leaves with eggs—second nymphs—were not collected since parasitoids from such leaves may not develop in the holding containers.

At the end of each collection day, all envelopes with leaf samples were transported to the insectary of the Hawaii Department of Agriculture, Plant Pest Control Branch in Honolulu. In the insectary, the leaves were carefully removed from the envelopes and placed in 70 mesh clear vinyl nylon screen collapsible lightweight aluminum cages (30 × 30 × 60 cm) appropriate for parasitic wasps′ emergence (https://www.bioquip.com (accessed on 20 October 2023)). Minute honey drops were placed on the inner surface of the mesh as food for adult wasps. We used smeared drops of SUE BEE^®^ SPUN^®^ honey on the inside cage sides and top for adult parasitoid feeding as indicated by HDOA insect rearing (https://siouxhoney.com/sue-bee-spun-honey/ (accessed on 20 July 2022)). Emerged adult parasitoids were aspirated and collected in falcon vials with tiny honey drops (Falcon 50 mL Conical Centrifuge Tubes; Fisher Scientific: A Thermo Fisher Scientific Brand (https://www.fishersci.com/shop/products/falcon-50ml-conical-centrifuge-tubes (accessed on 15 September 2023)).

Emerged parasitoids and remaining infested leaves were shipped to Greece and placed in the containment facility in the biosecurity greenhouse at the Scientific Directorate of Entomology and Agricultural Zoology, Benaki Phytopathological Institute (38°04′52.0″ N 23°48′47.9″ E). Emerged parasitoids were placed inside mesh cages with bitter orange *Citrus aurantium* L. (Rutaceae) seedlings infested by *A. spiniferus* in the biosecurity greenhouse under controlled environmental conditions of 25 °C ± 2 °C, R.H. 60–70%, and natural daylight photoperiod (daylight hours 6:40 a.m.–5:30 p.m.). *Citrus aurantium* seedlings were in mesh cages (45 × 45 × 45 cm), which were put inside larger ones (60 × 60 × 60 cm, 70 mesh). Polyester Chiffon white breathable fabric was used to cover cages for rearing delicate encyrtid-size parasitoids (https://www.moodfabrics.com/fashion-fabrics/polyester/chiffon (accessed on 10 October 2023)).

### 2.2. Identification of the Parasitoids

The identification of emergent parasitoids was based on morphological characteristics and molecular analyses. Morphological identification was performed according to appropriate keys and illustrations [20,21,22,23]. For distinguishing *Amitus hesperidum*: The female is shiny black (0.75 mm long). The female′s antenna is ten-segmented, with the last three segments forming a club. The male is like the female, with a filiform ten-segmented antenna and curved scape. A lateral plate-like process on the male fourth antennal segment is characteristic of the species, Figure 2D [20,21,23]. For identification of *Encarsia perplexa*, the mid lobe of the mesoscutum is dark, and T1 and T2 of the gaster are largely pale (Figure 2C). The male head is like that of the female. The mesosoma is orange–yellow except for the pronotum. The anterior half of the mid lobe of the mesoscutum, the propodeum, and the petiole are dark brown. The gaster is brown to dark (Figure 2C). Other characteristics for the differentiation of closely related species are explained in [22]. The specimens were prepared for slide mounting as described in slide preparation of chalcidoids by Noyes [24,25,26].

The insects that were destined for molecular analysis were stored in 1.5 mL microtubes with snap-cap RNase and DNase-free ClearLine^®^, filled with 98% ethanol (Analytical Grade, Fisher Scientific, Hampton, NH, USA). The specimens were stored and coded separately. In total, DNA was extracted from 45 parasitoids by using the DNeasy Blood and Tissue Kit (QIAGEN) according to the manufacturer’s protocol. At the beginning of the procedure, every single insect was left on a filter paper until the ethanol was completely removed and the final DNA volume reached 20 μL. Two sets of primers were used for species determination during the polymerase chain reaction, targeting the genes of Cytochrome Oxidase I (COI) and 28S ribosomal RNA, which can further investigate the species′ determination [27,28]. For the amplification of the barcoding gene, the primers LCO-1490 (5′-GGTCAACAAATCATAAAGATATTGG-3′) and HCO-2198 (5′-TAAACTTCAGGGTGACCAAAAAATCA-3′) were used [29], while the sets 28S-D2-F (5′-AGAGAGAGTTCAAGAGTACGTG-3′) and 28S-D2-R (5′-TTGGTCCGTGTTTCAAGACGGG-3′) were used for the amplification of the 28S gene [30].

Each PCR reaction mixture for both primer sets contained 5 µL of 10× PCR buffer, 1.5 µL of MgCl_2_ (50 mM), 0.5 µL of dNTPs (10 mM), 1 µL of each primer (10 µM), 5 µL of template DNA (20–40 ng), 0.5 μL of the thermostable Taq DNA polymerase (Platinum, Invitrogen), and molecular-grade water (up to 50 µL). The thermocycling program included an initial denaturation step of 3 min at 94 °C, followed by 35 cycles of 94 °C for 30 s, 51 °C (LCO-HCO) or 58 °C (28S) for 45 s, and 72 °C for 1 min, and a final step of extension at 72 °C for 5 min.

The two template amplifications were confirmed separately by using 5 μL of the PCR products on 1.2% agarose gel electrophoresis, which finally resulted in the observation of an expected length product of 658 bp and 550 bp, respectively. The rest of the volume of 45 μL was purified according to the supplier’s instructions for the NucleoFast 96 PCR Clean-up kit (Macherey-Nagel GmbH & Co. KG, Düren, Germany) and then forwarded to Macrogen Europe (Netherlands) for automated sequencing analysis. The obtained sequencing results were optimized, generated, and aligned through the software Geneious Prime 2023.0.1 (https://www.geneious.com/ (accessed on 10 June 2023)). The produced sequences were checked for their authenticity at the genus or species level according to the BLAST public interface of the National Center for Biotechnology Information (NCBI—https://blast.ncbi.nlm.nih.gov (accessed on 10 June 2023).

### 2.3. Infestation and Parasitism Rate

The parasitism rate of *Amitus* sp. and *Encarsia* sp. to *A. woglumi* from the collected leaves was estimated from the exit holes on the body of the nymphs. The exit hole of *A. woglumi* adults differs from the exit hole of the parasitoids by the shape of exit holes on the shells. The holes due to the exit of the parasitoids are circular and placed at the back of the nymph, while those from the exit hole of *A. woglumi* adults are T-shaped slits (Figure 2A,E). To determine the parasitism rate, all nymphs on each leaf were counted under a Nikon SMZ-745 stereomicroscope in the laboratory. The nymphs were listed as “parasitized″ if the shells had parasitoid circular exit holes, as in Figure 2E. All the emerged parasitoids were *Encarsia perplexa*. The unparasitized nymphs had the T-shape exit slit of the *A. woglumi*, as in Figure 2A, while broken shells were excluded from the results. Finally, the leaf samples were frozen at −20 °C for 72 h before being discarded.

The level of infestation was determined by the population size of the citrus blackfly, which was categorized depending on the total number of nymphs per leaf and the total number of infested leaves collected per location. The infestation was scored as follows: 1 = 1–10 nymphs, 2 = 11–30 nymphs, 3 = 31–99 nymphs, and 4 = ≥100 nymphs per leaf. An average score was calculated for each location. Since we collected only infested leaves, there was no 0 score. The eggs of the citrus blackfly were excluded from the score.

The maps were made using ArcGIS Pro, Version 3.0.3 (Redlands, CA: Environmental Systems Research Institute, Inc., https://www.esri.com (accessed on 17 July 2023), ESRI, and Natural Earth (free vector and raster map data @ naturalearthdata.com).

## 3. Results and Discussion

Survey results indicated that *A. woglumi* populations are rare in the Hawaiian Islands. Out of the 62 locations inspected for *A. woglumi* populations, its presence was found in only 11 sites (Figure 1). On Kauai Island, no infestation was found, and on Hawaii Island, *A. woglumi* populations were found in just 3 sites out of the 23 inspected. Moreover, the infestation rate was quite low (level of infestation 1.37–3.14), and only isolated citrus trees either on Oahu or in the Hawaii Islands were found with a large number of infested leaves (≥10 leaves/tree). Throughout the survey, populations of other Aleyrodidae were very low, and only a few isolated populations of *Aleurothrixus floccosus* (Maskell) and *Aleurodicus dispersus* Russell were found.

Parasitism of *A. woglumi* nymphs was much higher on Oahu than on Hawaii Island. Parasitism rates ranged from 0 to 28% on Hawaii Island and 11 to 65% on Oahu Island (Table 2 and Table 3).

Based on the morphological and molecular data, the emerging parasitoids were identified as *E. perplexa*. A few emerging parasitoids were identified as *Amitus spiniferus* Brethes, a parasitoid introduced to Hawaii for *A. floccosus*.

Specifically, the molecular comparison via the NCBI (National Center for Biotechnology Information) database resulted in high similarity rates, suggesting that the analyzed parasitoids undoubtedly belong to the genus *Encarsia*. Most of the consensus sequence produced belongs to *A. hesperidum*. Our survey results were similar to those of *E. perplexa*, at a range of 98.6 to 99.5%. The NCBI contained only limited data for the genus *Amitus* for both genes, a fact that limited the determination of some individuals to the genus level. Sequencing data provided by the USDA supports the claim that these parasitoids are highly similar to the species *Amitus spiniferus*, much more so than to *A. hesperidum*.

Our survey results confirm that *A. woglumi* on the Hawaiian Islands is effectively controlled by the introduced natural enemies and that *E. perplexa* is the dominant parasitoid species [2]. We were not able to find *A. hesperidum* either due to competition with *E. perplexa* or because its population is at very low levels, not being detectable with our sampling effort during this short survey. In other locations where both *E. perplexa* and *A. hesperidum* have been released to control *A. woglumi*, soon after the reduction in the infestation, *E. perplexa* had become the dominant species. This is mainly due to its longer lifespan compared to *A. hesperidum*. The latter is considered an effective parasitoid at high densities of its host, whereas *E. perplexa* is more efficient at low host densities [11].

In 1997, the infestation by *A. woglumi* was so severe on Oahu Island that every citrus tree, including oranges, lemons, and pummelo, had a range of 50–>600 citrus blackfly pupae/leaf (Figure 2A). In 1998, all citrus trees on Oahu (urban trees and orchards) had been reduced to no fruiting from the high infestation. There were complaints from citrus growers on the Hawaii and Kauai Islands requesting parasitoid introductions. Recently, there have been no reports of damage incurred by the pest or reports of growers or stakeholders having issues with citrus blackflies.

Non-citrus trees and ornamental shrubs (orange jasmine, *Murraya paniculate* (L.), Rutaceae) were also utilized for oviposition but were not normally affected by this whitefly. The mean number of egg masses/leaflet was 16.4 ± 2.8 (*n* = 25, during June 1997, Oahu Island) and 22.4 ± 3.1 eggs/mass (*n* = 22). Citrus blackfly eggs hatched with no development beyond the first nymph on non-citrus trees.

The infestation in 1997 was the worst on lemon trees (419.4 ± 32.4 pupae/leaf). Pummelo *Citrus maxima* (Burm.) Merr., the infestation was even higher with 112.5 ± 14.0 nymphs/cm^2^ leaf, compared with Meyer lemon, *Citrus meyeri* Yu. Tanaka (45.8 ± 15.0 nymphs/cm^2^ leaf), pink tecoma, *Tabebuia rosea* DC., Bignoniaceae (13.8 ± 2.1 nymphs/cm^2^ leaf), and mango (5.5 ± 1.3 nymphs/cm^2^ leaf). These records demonstrated the potential of the citrus blackfly to infest the Hawaiian citrus and mango trees even faster than any other mainland state because of the favorite environmental conditions [2] and unpublished data.

From the adult parasitoids that emerged either in Hawaii and shipped into Greece or emerged as adults in the quarantine facility of Benaki Phytopathological Institute in Athens, >3000 adults were introduced into rearing cages with young *Citrus aurantium* plants infested with *A. spiniferus*. Parasitoids were introduced into the cages as they emerged from *A. woglumi* nymphs for a period of almost 20 days. Only *E. perplexa* adults emerged, and they did not parasitize *A. spiniferus* in Greece. Parasitoids remain alive even after 30 days without parasitizing any *A. spiniferus* nymphs. *Encarsia perplexa* is known to parasitize *A. woglumi*, *Aleuroclava kuwani* (Takahashi), and *Aleuroplatus pectiniferus* Quaintance and Baker [22]. *Amitus hesperidum* was not recovered from the shipments. The parasitoid *A. hesperidum* was difficult to record even soon after its initial release on the Hawaiian Islands due to extremely low population densities, which was also confirmed by the current surveys. *Encarsia smithi*, a known parasitoid of *A. spiniferus*, has been found in the past on Oahu Island [16] but was not recovered during the current surveys.

The current study confirms that *A. woglumi* is efficiently controlled by natural enemies on the Hawaiian Islands and that the dominant parasitoid is *E. perplexa*. *Amitus hesperidum* is either extinct or may have been present in extremely low populations due to the very low host density and scattered populations of *A. woglumi*. That parasitoid has a short adult life span, which limits its searching efficiency for new suitable hosts to parasitize. According to the results of the surveys and the laboratory experiments, we can conclude that the parasitoid *E. perplexa,* which was recovered on the Hawaiian Islands, is able to control *A. woglumi* but is not the appropriate species for *A. spiniferus*. Additional exploratory surveys are ongoing in Southeast Asia, aiming to locate *A. hesperidum* or other *Encarsia* species that are reported to parasitize the orange spiny whitefly.

## Figures and Tables

**Figure 1 insects-14-00858-f001:**
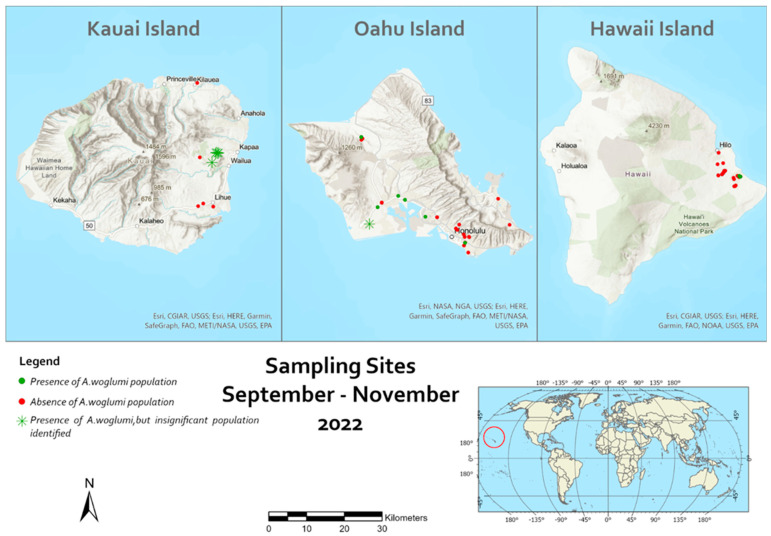
Sampling locations for *Aleurocanthus woglumi* and its parasitoids in the Hawaiian Islands (red circle). Sampling locations with GPS coordinates are shown in Table 2 and Table 3.

**Figure 2 insects-14-00858-f002:**
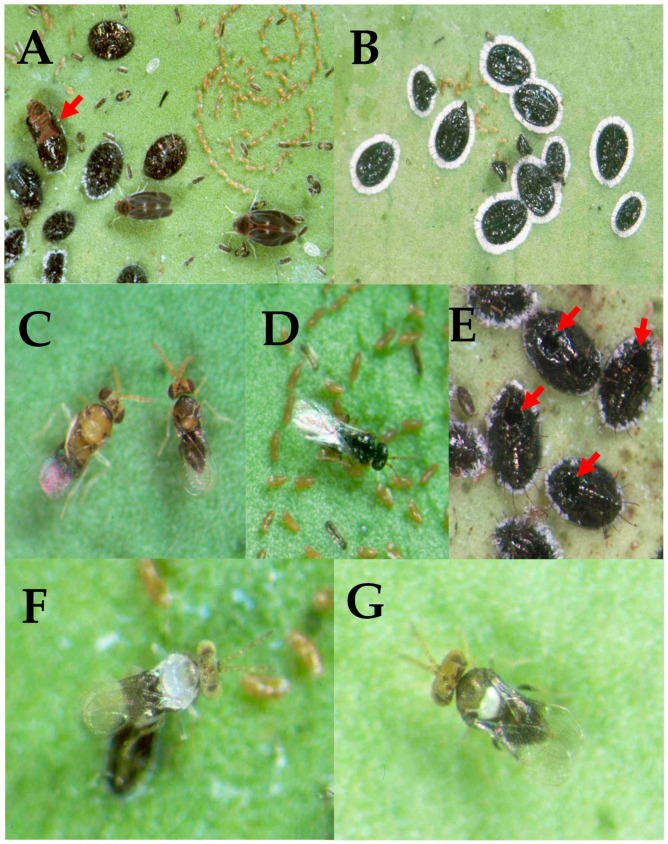
(**A**) Citrus blackflies, *Alerocanthus woglumi* (arrow showing unparasitized pupa and egg spirals); (**B**) orange spiny whiteflies, *A. spiniferus*, and their parasitoid assembly established in Hawaii. The pupal stage of the orange spiny whitefly with the white wax fringe that surrounds its margins is twice as large as the orange spiny whitefly; (**C**) *Encaria perplexa*, male and female, introduced from Guatemala in 1998; (**D**) *Amitus hesperidum*, introduced from Guatemala in 1998; (**E**) Citrus blackfly parasitized pupae, arrows showing parasitoid′s circular exit holes. The oval-shaped pupa is black and convex and has dark dorsal spines. The marginal wax tubes produce a cottony fringe on the pupal margin; (**F**) *Encarsia nipponica* adventive; (**G**) *Encarsia smithi*, introduced from Japan in 1975 and established on Oahu.

**Table 1 insects-14-00858-t001:** Hymenopterous parasitoid assembly of *Aleurocanthus spiniferus* and *A. woglumi* confirmed on the Hawaiian Islands, 1974–2002.

Target Whitefly (Hemiptera: Aleyrodidiae)	Introduced and Adventive Natural Enemies						
	Family	Species	Origin of Introduction	Year of Introduction and Release Period	Colonization Records (Total Numbers Released Per Island)	Establishment Records	Release Records
*Aleurocanthus* *spiniferus* (Quaintance), citrus orange spiny whitefly	Platygastridae	*Amitus* *hesperidum* Silvestri	Mexico	1974		Recorded on Oahu	Only Oahu Island
	Platygastridae	*Amitus* *spiniferus* (Brethes)	California	1979–1982	Oahu (200)	Introduced for *Aleurothrixus* *floccosus* (Maskell). Established on Oahu Island.	Only on Oahu Island in 1982.
	Aphelinidae	*Cales* *noacki* Howard	California	February 1981–August 1982	Lanai (275) Molokai (300) Oahu (12,149)	Introduced for *Aleurothrixus* *floccosus* (Maskell) Hawaii 1992 Kauai 1997 Oahu 1982	Hawaii 1992 Lanai 1982 Molokai 1982 Oahu 1982
	Aphelinidae	*Encarsia* *clypealis* (Silvestri)	Texas	1975		No recovery	Only on Oahu Island.
	Aphelinidae	*Encarsia* *perplexa* Huang and Polaszek	Texas	1975	Oahu (165)	No recovery	Only on Oahu, released as *Encarsia opulenta* (Silvestri).
	Aphelinidae	*Encarsia* *smithi* (Silvestri)	Japan (Nagasaki) and Guam	September 1974	Oahu (5330)	Established on Oahu September 1975, reported 1981, 1984, 1999 on Oahu.	Only on Oahu Island, some of the colonies from Guam in 1984 record (25 adults).
*Aleurocanthus woglumi* Ashby, citrus blackfly	Platygastridae	*Amitus* *hesperidum* Silvestri	Guatemala	Introduced 1998, released May 1999 –August 2000	Hawaii (20), Kauai (112), Maui (82), Oahu (8595)	Established	Hawaii, Kauai, Maui, Molokai, Oahu Islands
	Aphelinidae	*Cales* *noacki* Howard	California	February 1981–August 1982	Molokai (300) Oahu (12,149)	Introduced for *Aleurothrixus* *floccosus* (Maskell)	Only on Oahu Island in 1982. Recorded established on Oahu Island.
	Aphelinidae	*Encarsia* *perplexa* Huang and Polaszek	Guatemala	Introduced 1998, released April 1999–June 2002	Hawaii (49,190) Kauai (17,090) Maui (47,025) Molokai (27,165) Oahu (5740)	Established	Hawaii, Kauai, Maui, Molokai, Oahu Islands. First identified as *Encarsia opulenta* (Silvestri)
	Aphelinidae	*Encarsia nipponica* Silvestri	Adventive, native to Japan and China	-	-	Hawaii 2000, Kauai 2001 Oahu 1997, 1999	Fortuitous species
	Aphelinidae	*Encarsia smithi* (Silvestri)	Japan (Nagasaki) and Guam	September 1974	Reported on Oahu 1989, 1999 on citrus	Established on Oahu September 1975 Reported in 1999 on Oahu	Only on Oahu Island.

**Table 2 insects-14-00858-t002:** Infestation rates and parasitization of *A. woglumi* infesting Citrus species on Oahu Island during September–November 2022. **^a^** residential house, roadside, highway service area, farmland, research station, community garden, cemetery, park, and church.

Survey Area and Habitat ^a^	Citrus Plant	Coordinate and Elevation	Date	Total Number of Leaves Collected	Total Number of Nymphs	Level of Infestation	% Parasitism
University of Hawaii at Manoa	Sour orange	21°18′17.98″ N 157°48′51.87″ W, 40 m	26 September 2022	0	0	0	-
University of Hawaii at Manoa	Sweet lime	21°18′15.16″ N 157°48′48.99″ W, 38 m	“	0	0	0	-
Ala Wai Community Garden	Lime	21°17′02.50″ N 157°49′37.15″ W, 1.5 m	“	0	0	0	-
Waimanalo Research Station	Citron,	21°20′07.91″ N 157°42′55.01″ W, 23 m	“	0	0	0	-
Kapaakea Lane, Honolulu	Tangerine	21°17′27.82″ N 157°49′25.95″ W, 4 m	“	0	0	0	-
Kapaakea Lane, Honolulu	Pummelo	21°17′25.55″ N 157°49′26.61″ W, 3.6 m	“	9	118	1.44 ± 0.49	17.8%
Pearl City Urban Botanical Garden	Sweet orange	21°23′38.76″ N 157°58′38.08″ W, 8.5 m	27 September 2022	71	1471	1.76 ± 0.78	56.6%
Diamond Head Community Garden	Tangelo	21°16′02.09″ N 157°58′59.55″ W, 0.9 m	“	0	0	0	-
Moanalua Gardens	Grapefruit	21°20′52.68″ N 157°53′28.74″ W, 6.4 m	“	0	0	0	-
North Shore. Poamoho Research Station	Sweet orange	21°32′38.00″ N 158°05′16.28″ W, 189 m	28 September 2022	345	11,981	2.17 ± 0.95	19.7%
Salt Lake Blvd	Sweet lime	21°21′09.13″ N 157°55′30.36″ W, 29 m	“	75	158	1.37 ± 0.48	53.8%
North Shore, Hawaii Queen bees, Hinshaw Farms	Sour orange	21°32′11.10″ N 158°05′16.82″ W, 226 m	“	0	0	0	-
North Shore, Hawaii Queen bees, Hinshaw Farms	Lemon	21°32′17.54″ N 158°05′14.90″ W, 225 m	“	0	0	0	-
Hawaii Agriculture Research Center, Kunia	Tangelo	21°23′09.02″ N 158°02′14.67″ W, 82.6 m	2 October 2022	0	0	0	-
Hawaii Agriculture Research Center 2	Lemon	21°23′07.28″ N 158°02′17.16″ W, 84.4 m	“	0	0	0	-
Manoa valley community gardens	Citron	21°18′53.77″ N 157°48′25.49″ W, 57 m	“	0	0	0	-
Nuuanu Mauna ala Royal mausoleum	Pummelo	21°19′30.33″ N 157°50′49.81″ W, 64 m	26 October 2022	0	0	0	-
Nuuanu old Pali drive	Lemon	21°21′14.24″ N 157°48′37.68″ W, 308 m	“	0	0	0	-
Tantalus lookout	Tangerine	21°19′06.92″ N 157°49′48.23″ W, 277 m	“	0	0	0	-
Kailua	Lemon	21°23′47.09″ N 157°44′44.49″ W, 0.6 m	“	0	0	0	-
Mahiole street, Moanaloa	Sour orange	21°20′49.79″ N 157°53′21.81″ W, 11 m	31 October 2022	0	0	0	-
North Shore Poamoho Research Station	Sweet orange and sweet lime	21°32′38.00″ N 158°05′16.28″ W, 189 m	“	573	23,168	2.34 ± 0.81	24.9%
Waikele	Tangerine	21°24′03.18″ N 158°00′13.75″ W, 60.6 m	1 November 2022	13	389	2.00 ± 1.03	34.4%
Aloun farm	Pummelo	21°22′28.47″ N 158°02′42.94″ W, 56.7 m	“	14	212	1.78 ± 0.67	11.5%
Royal Mausoleum of Hawaii	Lime	21°19′30.33″ N 157°50′49.81″ W, 64 m	“	0	0	0	-

**Table 3 insects-14-00858-t003:** Infestation rates and parasitization of *A. woglumi* infesting Citrus species on Hawaii Island during October 2022. **^a^** residential house, roadside, highway service area, farmland, research station, community garden, cemetery, park, and church.

Survey Area and Habitat ^a^	Citrus Plant	GPS and Elevation	Date	Total Number of Leaves Collected	Total Number of Nymphs	Level of Infestation	% Parasitism
Waiakea	Sweet orange	19°38′30.98″ N 155°04′51.67″ W, 194 m	27 October 2022	0	0	0	-
Waiakea Research station	Lime	19°38′41.46″ N 155°04′37.82″ W, 173 m	“	0	0	0	-
Hawaii Department of Agriculture, Hilo	Lemon	19°42′23.03″ N 155°04′26.25″ W, 11 m	“	0	0	0	-
Kurtistown 1	Pummelo	19°35′35.98″ N 155°03′30.41″ W, 206 m	“	0	0	0	-
Kurtistown 2	Pummelo	19°35′30.36″ N 155°03′29.91″ W, 199 m	“	0	0	0	-
Kurtistown 3	Sweet orange	19°35′00.29″ N 155°03′36.53″ W, 242 m	“	0	0	0	-
Kurtistown 4	Sour orange	19°34′38.83″ N 155°03′56.19″ W, 271 m	“	0	0	0	-
Hawaiian Paradise Park 1	Grapefruit	19°36′12.88″ N 154°56′51.46″ W, 13 m	28 October 2022	173	17,848	3.14 ± 0.76	2.7%
Hawaiian Paradise Park 2	Pummelo	19°36′12.88″ N 154°56′51.46″ W, 13 m	“	7	423	3.00 ± 0.50	0.0%
Hawaiian Paradise Park 3	Pummelo	19°34′17.70″ N 154°57′21.50″ W, 50 m	“	6	212	2.50 ± 0.80	28.3%
Hawaiian Paradise Park 4	Sweet orange	19°34′35.47″ N 154°57′18.39″ W, 9 m	“	0	0	0	-
Hawaiian Paradise Park 5	Citron	19°34′52.95″ N 154°57′39.69″ W, 39 m	29 October 2022	0	0	0	-
Hawaiian Paradise Park 6	Lime	19°34′14.46″ N 154°57′45.16″ W, 40 m	30 October 2022	0	0	0	-
Hawaiian Paradise Park 7	Lemon	19°35′39.96″ N 154°57′21.77″ W, 28 m	31 October 2022	0	0	0	-
Keeau 1	Sour orange	19°37′06.79″ N 155°02′37.40″ W, 121 m	29 October 2022	0	0	0	-
Keeau 2	Lemon	19°36′52.38″ N 155°02′48.90″ W, 149 m	“	0	0	0	-
Keeau 3	Tangerine	19°36′23.02″ N 155°02′59.69″ W, 177 m	“	0	0	0	-
Keeau 4	Sweet orange	19°37′11.96″ N 155°02′32.56″ W, 117 m	“	0	0	0	-
Keeau 5	Pummelo	19°37′42.04″ N 155°02′19.12″ W, 123 m	“	0	0	0	-
Keeau 6	Tangelo	19°38′42.04″ N 155°02′32.56″ W, 79 m	“	0	0	0	-
Ainaloa 1	Tangerine	19°31′40.76″ N 154°59′35.89″ W, 217 m	“	0	0	0	-
Ainaloa 2	Lime	19°31′20.12″ N 155°00′05.43″ W, 226 m	“	0	0	0	-
Ainaloa 3	Lemon	19°31′47.83″ N 154°59′49.05″ W, 201 m	“	0	0	0	-

## Data Availability

The data presented in this study are available on request from the corresponding author.

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
