# Peer review of "A Survey in Hawaii for Parasitoids of Citrus Whiteflies (Hemiptera: Aleyrodidae), for Introduction into Greece"

_insects, 2023, doi:10.3390/insects14110858_

Round 1
Reviewer 1 Report (Previous Reviewer 1)
Comments and Suggestions for Authors
survey in Hawaii for parasitoids of citrus whiteflies 1 (Hemiptera: Aleyrodidae), for introduction into Greece
Review comments
Line 42: Provide authors for each species.
L 61-62. Reword this sentence.
L 63-64: does not make sense.
L 66: It should be colon :, not.
L 67: Delete; too many in keyword section.
L 145: provide names and GPS coordinates of islands
L 145: give information about host plants-age, size, etc.
L 160: Reword this sentence.
L 163: delete `since--------parasitods.’
L 169: Provide source of aluminium cages
L 176: Did not honey drop trap parasitoid adults?
L 187: Provide daylight hours.
L 189: Please specify. Very important to know.
L 197: Provide sampling locations in the form of a table showing region, location with GPS coordinates.
L 203: Use `characters’ instead of `examination’.
L 204: Provide reference for keys.
L 260: Incomplete sentence.
L 262- 265: Reword and organize. Difiicult to understand.
L 266: This needs to be checked for accuracy.
L 294: Show infestation rate in parenthesis
L 296: ≥ 10 leaves)--per location or per tree?
L 297: name what are the others Aleyrodidae?
L 304: ` Only were------floccosus.’--rewrite this sentence correctly.
L 306: Provide full names of all acronyms.
L 314: ` A. woglumi on the Hawaiian Islands is effectively controlled by the introduced natural enemies’-- What is the basis of this conclusion.
L 393: Fix references nos. 1, 2, 3, 13, 24.

In my comments, i used line numbers to show unclearity in the expression. of meaning.
Author Response
Response to Reviewer 1:
Comments and Suggestions for Authors
survey in Hawaii for parasitoids of citrus whiteflies 1 (Hemiptera: Aleyrodidae), for introduction into Greece
Many thanks to the anonymous reviewer for his edits and careful revision that improved the manuscript. All corrections were considered, made in blue in the revised version.
Review comments
Line 42: Provide authors for each species.
Response: Thank you, corrected.
L 61-62. Reword this sentence.
Response: Sentence reworded in the revised text “The dominant parasitoid was characterized as Encarsia perplexa, using molecular analysis. Its parasitism rates ranged 0 –28 % on the island of Hawaii and 11 – 65% on the island of Oahu”.
L 63-64: does not make sense.
Response: Sentence reworded in the revised text “Emerged parasitoids have been reared in Greece for evaluations”.
L 66: It should be colon :, not.
Response: Thank you, corrected.
L 67: Delete; too many in keyword section.
Response: Aleurocanthus woglumi deleted.
L 145: provide names and GPS coordinates of islands
Response: Names and GPS coordinates of islands were inserted “ Hawaii Island with the GPS coordinates of 19° 44' 30.3180'' N , 155° 50' 39.9732'' W; Oahu Island GPS coordinates of 21° 18' 56.1708'' N, 157° 51' 29.1348'' W; Kauai Island GPS coordinates of 22° 6' 30.7548'' N, 159° 29' 48.3540'' W (https://www.latlong.net)”.
L 145: give information about host plants-age, size, etc.
Response: The suggested information on Host plants names, survey area and habitat, number of leaves collected are presented in Tables 2,3.
L 160: Reword this sentence.
Response: Sentence reworded as “Infested leaves were picked according to the stage of nymphs of the citrus blackfly”.
L 163: delete `since--------parasitods.’
Response: Sentence deleted ( since these are the stages contain developed parasitoids).
L 169: Provide source of aluminium cages
Response: The source of aluminium cages added to text “ https://www.bioquip.com”.
L 176: Did not honey drop trap parasitoid adults?
Response: Thank you we added “the vials with tiny honey drops” for clarity.
L 187: Provide daylight hours.
Response: added to text “daylight hours 6: 40 AM – 5:30 PM”.
L 189: Please specify. Very important to know.
Response: Thank you. This statement added to text for clarity “Polyester Chiffon white breathable fabric used to cover cages for rearing delicate encyrtid size parasitoids (https://www.moodfabrics.com/fashion-fabrics/polyester/chiffon)”.
L 197: Provide sampling locations in the form of a table showing region, location with GPS coordinates.
Response: Thank you, sampling locations with GPS coordinates are shown in Table 2, 3.
L 203: Use `characters’ instead of `examination’.
Response: Thank you, corrected.
L 204: Provide reference for keys.
Response: The appropriate keys and illustrations are cited in four references [20, 21, 22, 23]. In addition, our explanation for reader.
L 260: Incomplete sentence.
Response: sentences revised to “nymphs were listed as “parasitized” if the shells had parasitoid circular exit holes as in Figure 2 E. All eclosed parasitoids were Encarsia perplexa. The un-parasitized nymphs had the T-shape exit slit of the A. woglumi as in Figure 2 A, while broken shells were excluded from the results”.
L 262- 265: Reword and organize. Difiicult to understand.
Response: Sentences reworded “The nymphs were listed as “parasitized” if the shells had parasitoid circular exit holes as in Figure 2 E. All eclosed parasitoids were Encarsia perplexa. The unparasitized nymphs had the T-shape exit slit of the A. woglumi as in Figure 2 A, while broken shells were excluded from the results.
L 266: This needs to be checked for accuracy.
Response: sentences reworded “The level of infestation was determined by the population size of the citrus blackfly, which was categorized depending on the total number of nymphs per leaf and the total number of infested leaves collected per location. The infestation was scored as follows: 1 = 1 – 10 nymphs, 2 = 11 – 30 nymphs, 3 = 31 – 99 nymphs, 4 = ≥ 100 nymphs per leaf”.
L 294: Show infestation rate in parenthesis
Response: Thank you. Infestation rate added in parenthesis as “Moreover, the infestation rate was quite low (level of infestation 1.37 – 3.14)”.
L 296: ≥ 10 leaves)--per location or per tree?
Response: Islands were found with large number of infested leaves (≥ 10 leaves/tree).
L 297: name what are the others Aleyrodidae?
Response: The following sentence in text “Throughout the survey, populations of other Aleyrodidae were very low and only a few isolated populations of Aleurothrixus floccosus (Maskell), and Aleurodicus dispersus Russell, were found”.
L 304: ` Only were------floccosus.’--rewrite this sentence correctly.
Response: sentence reworded “A few emerged parasitoids were identified as Amitus spiniferus Brethes, a parasitoid introduced to Hawaii for A. floccosus”.
L 306: Provide full names of all acronyms.
Response: National Center for Biotechnology Information (NCBI)
Hawaii Department of Agriculture (HDOA)
L 314: ` A. woglumi on the Hawaiian Islands is effectively controlled by the introduced natural enemies’-- What is the basis of this conclusion.
Response: A reference added to the text. The conclusion is based on a review article on the success of biocontrol introductions in Hawaii. Ramadan, M., M., Kaufman, L., V., Wright, M., G. 2023. Insect and weed biological con-trol in Hawaii: Recent case studies and trends. Biol. Control. 2023, 179, 105170. https://doi.org/https://doi.org/10.1016/j.biocontrol.2023.105170
L 393: Fix references nos. 1, 2, 3, 13, 24.
Response: Thank you, all references are now fixed to the journal style.
Reviewer 2 Report (Previous Reviewer 3)
Comments and Suggestions for Authors
Thanks the authors for the work carried out. The changes made to the manuscript have made it clearer also from the point of view of the description of the materials and methods which in the previous version seemed less detailed. The purpose of the work and the situation of the monitoring carried out are also clearer.
Author Response
Reviewer 2:
Thanks the authors for the work carried out. The changes made to the manuscript have made it clearer also from the point of view of the description of the materials and methods which in the previous version seemed less detailed. The purpose of the work and the situation of the monitoring carried out are also clearer.
Response Many thanks to anonymous reviewer for reading and his careful editing of the manuscript.
This manuscript is a resubmission of an earlier submission. The following is a list of the peer review reports and author responses from that submission.
Round 1
Reviewer 1 Report
Comments and Suggestions for Authors
The manuscript on `A survey in Hawaii for parasitoids of citurs --------into Greece’ contains good idea to introduce biological control citrus blackfly. I am fully in favor of this work. However, I have some questions about various aspects of conducting this study. I will suggest authors put some efforts to improve the materials and methods section. I am surprised that the work has been done without replicating any trials. For example, authors went to collect samples and picked infested leaves from a plant. They could use the tree as a field and make different plots to collect sample. This way they could assess severity of pest and parasitoid distribution and abundance. However, I wish authors to think and reorganize the manuscript. I vote for accepting this manuscript with major changes. I included my comments directly in the manuscript for authors to act.
Line 119: There is no information about rearing.
Lines 120-122: Please indicate how these sites were selected. Were these sites selected based on distance or otherwise?
Line 124: Provide description and source of Paper envelopes.
Line 125: Did you mean any development stage? If so, what stage was selected.
Line 126: How did you recognize older nymphs?
Line 130: Provide size of the mesh and the cage.
Line 131: Pl. mention the source of honey. How about the composition of honey?
Line 133: You should clearly declare the source and size. I know, it is not important to you. But the information will help readers like us.
Line 141. You may want to show mesh size.
Line 150. Provide some key characters for morphological identification. Readers want to know at the time they read the manuscript.
Line 150. Elaborate by describing important aspects of slide mounting. That helps readers and specific researchers a lot.
Line 152: Is that all about storing specimens? It is important to provide size and source of the tubes and ethanol. Did you place all specimens in the same tube or separate tubes?
Line 191. The following statement is not clear about determining parasitism rate. Please provide the methods.
Line 192. Provide source of the stereomicroscope.
Line 224. Delete.
Line 224. Provide the number within parenthesis.
Line 264. Please erase this column (Date). Only two dates were used which can be mentioned in the text.
Line 279-288. There is no proof of this data. The information should be somewhere before citing. Even in the form of presentation is good.
Line 292. Mention age at the beginning.
Line 305. What is the basis of claiming this.
Comments on the Quality of English LanguageEnglish language quality is ok.
Author Response
Thank you very much for anonymous reviewers' comments. All suggested notes were considered, this improved the readability of our manuscript.

Reviewer 2 Report
Comments and Suggestions for Authors
Title: A survey in Hawaii for parasitoids of citrus blackfly, Aleurocanthus woglumi (Hemiptera: Aleyrodidae), for introduction into Greece
General comments:
In the manuscript the authors report the results of field collections of whitefly infested leaves in search for parasitoids.
The manuscript has several issues, it should be reconstructed to be much more focused. The basic concept of the study is also rather confusing, the authors were seeking for the parasitoids of a whitefly species Aleurocanthus woglumi in Hawaii, for introduction of the natural enemies to Greece, however, as the authors pointed out, the pest has not been detected in Europe so far. Furthermore, the authors tested the dominant parasitoid of A. woglumi against another pest Aleurocanthus spiniferus in Greece in a quarantine facility with no success. Although the authors stated that finding a potent parasitoid against A. spiniferus was the main goal of the study, the whole manuscript focuses on A. woglumi as it is also emphasized in the title.
Beside these there are also considerable mismatches in the general structure of the manuscript e.g. methods described in Results and Discussion.
As a conclusion I recommend the manuscript to be reconstructed so that aims and achievements could be made much more understandable to the readers.
Comments on the Quality of English LanguageThe text of the manuscipt requires some language editing.
Author Response
Thank you for reviewer 2 suggested notes, all were considered in the revised manuscript.

Reviewer 3 Report
Comments and Suggestions for Authors
The manuscript deals with a topic of great interest: the introduction and adaptation to new environments of parasitoids for the biological control of alien pests. The monitoring carried out in the Hawaiian Islands in relation to the spread of parasitoids introduced at different times for the control of populations of one of the most harmful whiteflies to Citrus sp. crops is very important, but the shortness of the survey period and some methodological gaps make the manuscript to be reviewed completely. Although there were many survey sites, the two-month analysis alone seems to be too short compared to the conclusions reached by the authors. Also, the evaluation of the percentages of parasitization using the counting of the exit holes, is not very effective. In this way a percentage of general parasitization is obtained, but, in this way a false percentage of parasitization is obtained, since the individuals of the parasitoids still inside the nymphs or puparia are not considered. Furthermore, the prevalence of the most widespread parasitoid species (Encarsia perplexa) is obtained only a posteriori from the examination of the adults that have emerged in the laboratory in a very short period: also these data could be distorted. Perhaps this observation should be defined at the time of the monitoring and not made of it as an absolute result. Furthermore, the integration that emerges from the title between the monitoring done in Hawaii and what could happen in Greece is very forced and not well explained and the conclusions in this case seem poorly fitting. The results obtained in the Greek facility is very interesting as the E. perplexa did not parasitized A. spiniferus species. In this regard, it is not clear why, where both species of Aleurocanthus appear to be present in Hawaii, it was decided to monitor only A.woglumi, when the species to be monitored and control in Greece is A. spiniferus. So, on the one hand, therefore, a certain weakness is noted in the monitoring activity and on the other, the second part of the experimental activity carried out in Greece is not considered sufficiently explained and the conclusions seem weakly consistent. A complete revision of the manuscript is recommended to give a better approach that takes into consideration a snapshot, referring to the short period of study, of the situation concerning the parasitization of A. woglumi in the different sites in Hawaii. It will be necessary to illustrate in a more sufficient way the reason for choosing only this species of aleyrodid, instead of both, given that the second part of the work in Greece then takes into consideration A. spiniferus as a target. Addressing the experimental test carried out in Greece, I would also make a brief introduction by saying which studies are necessary for the introduction of new species for control, emphasizing that, in any case, what has been done is only a very preliminary study.
Specific comments
Introduction
Line 97: there is a mistake in the name of the species Aleurothrixus
Line 107-113: here you could explain why you have chosen only A. woglumi and not the two species.
Line 116: here you can better explain what should be the studies to approach the introduction of an exotic organism for biological control and that only a preliminary study has been done in Greece in the present work
Line 187: the exit hole from the nymphs is a nonspecific sign for a particular species, explain it
Line 231: explain what kind of parasitization you have calculated, in relation to the method used and the short term
Line 265-287: references are missing or these are your data? Or Ramadan unpublished data?
Figure 2: C letter Encarsia; G what about the description of pupal stage when there is a picture of an adult of E. smithi?
Author Response
Thank you for reviewer 3 suggested notes, all were considered in the revised manuscript.
